Tae-miR396b regulates TaGRFs in spikes of three wheat spike mutants

Yao Ziping 1
Wang Qi 1
Xue Ying 1
Liang Zhiheng 1
Ni Yongjing 2
Jiang Yumei 1
Zhang Peipei 1
Wang Ting 1
Li Qiaoyun 1
Li Lei 1
Niu Jishan 1 jsniu@henau.edu.cn
1 National Centre of Engineering and Technological Research for Wheat, Henan Agricultural University , Zhengzhou, Henan , China
2 Henan Engineering Research Center of Wheat Spring Freeze Injury Identification, Shangqiu Academy of Agricultural and Forestry Sciences , Shangqiu , China
Nogueira Fabio
Electronic publication date: 2024 Nov 22
Publication date: 2024
Volume: 12
Electronic Location ID: e18550
Received 2024 Jan 25; Accepted 2024 Oct 28
Copyright: © 2024 Yao et al.
Copyright year: 2024
Copyright holder: Yao et al.
License: This is an open access article distributed under the terms of the Creative Commons Attribution License, which permits unrestricted use, distribution, reproduction and adaptation in any medium and for any purpose provided that it is properly attributed. For attribution, the original author(s), title, publication source (PeerJ) and either DOI or URL of the article must be cited.
License URL: https://creativecommons.org/licenses/by/4.0/

Keywords: Wheat (Triticum aestivum L.), Spike, Mutant, Tae-miR396b, Growth regulating factor (GRF), Regulation

Funding: National Natural Science Foundation of China 32171972 Internal Foundation of National Key Laboratory of Crop Science on Wheat and Maize SKL2022ZZ06 This study was supported by National Natural Science Foundation of China (NSFC, 32171972), and Internal Foundation of National Key Laboratory of Crop Science on Wheat and Maize (SKL2022ZZ06). The funders had no role in study design, data collection and analysis, decision to publish, or preparation of the manuscript.

==============================
Tillering and spike differentiation are key agronomic traits for wheat (Triticum aestivum L.) production. Numerous studies have shown that miR396 and growth-regulating factor genes (GRFs) are involved in growth and development of different plant organs. Previously, we have reported that wheat miR396b (tae-miR396b) and their targets TaGRFs (T. aestivum GRFs) play important roles in regulating wheat tillering. This study was to investigate the regulatory roles of tae-miR396b and TaGRFs played during wheat spike development. Wheat cultivar Guomai 301 (wild type, WT) and its three sipke mutants dwarf round spike mutant (drs), apical spikelet sterility mutant (ass) and prematurely terminated spike differentiation mutant (ptsd1) were studied. Three homeologous genes of tae-miR396b on the long arms of chromosomes 6A, 6B, and 6D were identified, and they encoded the same mature miRNA. Complementary sequences of mature tae-miR396b were identified in 23 TaGRFs, indicating they were the target genes of tae-miR396b. Tae-miR396b had different regulatory effects on TaGRFs between Guomai 301 and its mutants. TaGRF2-7A was confirmed to be the target gene of tae-miR396b by molecular interaction assay. The expression levels of tae-miR396b and TaGRFs were different between WT and mutants drs, ass and ptsd1 at the floret primordium visible (S1), the two awns/spikelet reaching apical meristem of the spikelet (S2), and the green anther stage (S3). The expression level of tae-miR396b in WT was significantly higher than that in mutants drs and ass. The most TaGRFs were negatively regulated by tae-miR396b. The abnormal expressions of TaGRF1 (6A, 6D), TaGRF2 (7A, 7B, 7D), TaGRF4 (6A, 6B), TaGRF5 (4A, 7A, 7D), and TaGRF10 (6A, 6B, 6D) were important causes for abnormal spike development in the three mutants. This study laid foundation for further elucidating functions of tae-miR396b and TaGRFs underlying wheat spike development. Regulating tae-miR396b and TaGRFs will be a new approach for wheat high yield breeding.

Introduction

Wheat (Triticum aestivum L.) is one of the most important crops in the world. Tillering and spike differentiation are two determining traits for wheat production, so they are of significant scientific and economic interest (Sreenivasulu & Schnurbusch, 2012). Per plant yield of wheat consists of fertile spikes per plant, fertile spikelets per spike, grains per spike, and grain weight (Sreenivasulu & Schnurbusch, 2012). There are many factors affecting wheat yield, among which tillering and spike differentiation are two key traits (Ao et al., 2010). The differentiation process of wheat young spike is a critical stage to achieve large spike size and more grains (Li et al., 2019). Spike development and growth have a great effect on the growth and survival of florets and the number of grains (Jiao et al., 2019; Waddington, Cartwright & Wall, 1983). Heading time is an important trait in the transition from vegetative to reproductive growth of cereal crops, which affects the adaptability of crops to various environmental conditions (Murai et al., 2003). Too high tillering level will impact the number of fertile spikes, and ultimately leads to a decrease in wheat yield (Kebrom et al., 2012). Spike development directly affects the number of fertile spikelets and grains per spikelet. Researches on wheat spikes have benefited from the large number of spike mutants. They can help us to understand the key genes and molecular mechanisms behind floret and grain developments (Sreenivasulu & Schnurbusch, 2012).

MiRNA is a type of single stranded non coding RNA with a length of about 20–25 nt. MiRNA widely exists in animals, plants and some viruses and plays roles in RNA silencing and post transcriptional regulation of gene expression (Zhu et al., 2022). The first microRNA, miRNA-lin-4, was discovered in the 1990s (Lee, Feinbaum & Ambros, 1993). Since then, microRNAs have been studied in many species including crops. A wheat small RNA library was constructed and identified 58 microRNAs (miRNAs). Northern blot analysis indicated that several of the newly identified miRNAs were preferentially expressed in specific tissues of wheat. Additionally, a significant number of monocot-specific miRNAs were also identified. These findings suggest that both conserved miRNAs and those unique to wheat play crucial roles in physiological processes, including wheat growth, development, and stress response (Yao et al., 2007). Transcription factors participate in many important cellular processes, such as signal transduction, morphogenesis, and environmental stress responses, by influencing or controlling the expressions of some specific genes (Chen & Cao, 2014). Some transcription factor families are unique in plants, such as WRKY (Eulgem et al., 2000), R2R3-MYB (Stracke, Werber & Weisshaar, 2001), NAC (Olsen et al., 2005), TIFY (Vanholme et al., 2007), SBP-box (Yang et al., 2008) etc.

Growth-regulating factors (GRFs) are a kind of plant-specific transcription factors. GRFs play multiple important regulatory roles in the growth and development of plants. They are identified for their roles in stem and leaf development, flower and seed formation, root development, and the coordination of growth processes under adverse environmental conditions (Bao et al., 2014; Debernardi et al., 2014; Hewezi et al., 2012; Liang et al., 2013; van der Knaap, Kim & Kende, 2000; Wu et al., 2014). The expressions of several GRFs are controlled by miR396, and the miR396-GRFs regulatory module appears to be central in several of these processes (Baucher et al., 2012; Liu et al., 2014; Omidbakhshfard et al., 2015; Rodriguez et al., 2015, 2010; Wang et al., 2011). Obviously, miR396 and GRFs are two key gene expression regulation factors in plant species, however, wheat related studies are limited.

Previously, we found that the miR396-TaGRFs regulatory module played important roles in regulating wheat tillering (An et al., 2019; He et al., 2018; Zhang et al., 2021). However, their roles in wheat spike development are largely unknown. We obtained several spike mutants, including dwarf round spike mutant (drs), apical spikelet sterility mutant (ass) and prematurely terminated spike differentiation mutant (ptsd1) from the EMS-treated cultivar Guomai 301 (wild type, WT) (Jiao et al., 2019; Ni et al., 2015). In this study, we want to explore the roles of tae-miR396b-TaGRFs played in the abnormal wheat spike developments by analyzing the differential expressions of tae-miR396b and TaGRFs in WT and its mutants. Guomai 301 and its three mutants have the same genetic background, which is one advantage for study the roles of miR396-TaGRFs regulatory module as well as the effects of the mutated genes drs, ass and ptsd1. The results were reported here.

Materials and Methods

Plant materials and growth methods

Guomai 301 (WT) was bred in our laboratory, Henan Technology Innovation Centre of Wheat, Henan Agricultural University. The seeds of WT were treated with EMS and planted at Shuangba Experimental Station of Shangqiu Academy of Agricultural and Forestry Sciences (34°25′N, 115°39′E, 49 m above sea level). Where belongs to the continental monsoon climate zone. The seeds were treated with 0.4% EMS solution for 4 h, and then washed with tap water over night (Ni et al., 2015). The mutants drs, ass and ptsd1 with spike—type variation were obtained in the M2 generation in 2014. Since 2014, the mutants have been planted and observed continuously in Xingyang experimental field in Zhengzhou (34°86′N, 113°44′E, 114 m above sea level) and Shuangba experimental station in Shangqiu. The mutants were at M11 generation in 2023. The field experiments were carried out in a completely randomized design as described by us (Duan et al., 2015).

Identification and analysis of tae-miR396b

The miR396b sequences of wheat and other species were obtained by comparing and analyzing the miR396b sequences in miRBase (http://www.mirbase.org/) database. The tae-miR396b and their annotations were obtained by alignment analysis miR396b sequences in wheat EnsemblPlants (http://plants.ensembl.org/index.html) in wheat genome data. RNAfold software (http://unafold.rna.albany.edu/?q=mfold/RNA-Folding-Form) was used to predict the hairpin structures of tae-miR396b precursors. Sequence alignment analysis was carried out using DNAMAN 6.0 software (https://dnaman.software.informer.com/).

Prediction of the tae-miR396b target sites on TaGRFs

The sequence of tae-miR396b was downloaded from the miRNA database (http://www.mirbase.org/). The tae-miR396b binding sites of TaGRFs were predicted on psRNATarget website (https://www.zhaolab.org/psRNATarget/). Parameters were chosen as the system default values.

Tissue specific expression analysis of TaGRFs

The raw data of TaGRFs’ expressions in various tissues and organs during wheat growth and development were downloaded from the Wheat Expression Browser website (http://www.wheat-expression.com/). Six samples of roots, leaves and spikes of Chinese Spring were analyzed at the 2nd node detectable (GS32) and boots just visibly swollen (GS43). The definition of wheat developmental stages were referred to the description of Zadoks, Chang & Konzak (1974). Tissue specific expression heatmaps of TaGRFs were generated using TBtools software. Expression profiles were drawn based on transcripts per million (TPM) values of TaGRFs.

Real-time qRT-PCR

qRT-PCR was performed as described previously by us (An et al., 2019). The primers of TaGRFs were designed using Primer-Blast of NCBI website (https://www.ncbi.nlm.nih.gov/tools/primer-blast/). All the primer sequences were listed in Table S1. The actin gene was used as the internal control for real-time quantification of TaGRFs, and U6 was used as the internal control for real-time quantification of tae-miR396b. Each reaction was performed with three technical repeats. The relative expressions of TaGRFs and tae-miR396b were calculated by 2−ΔΔCt methods (Livak & Schmittgen, 2001).

Correlation analysis between tae-miR396b and TaGRFs expressions in WT and mutants

Based on the results of qRT-PCR, Origin (https://www.originlab.com/) software was used to analyze the Pearson’s correlations of the expression levels of tae-miR396b and TaGRFs in WT and the mutants ass, drs, and ptsd1. The developmental stages were defined referred to that described by Vahamidis et al. (2014). Samples at three stages of the floret primordium visible stage, the two awns/spikelet reaching apical meristem of the spikelet stage, and the green anther stage (Vahamidis et al., 2014) were used to study the regulatory effect of tae-miR396b on different TaGRFs.

Expression vector construction and interaction analysis of tae-miR396b and TaGRF2-7A

According to the precursor sequence of tae-miR396b, the primers with SpeI restriction site were designed for cloning tae-miR396b and construction the expression vector pCAMBIA1304-tae-miR396b (Table S1). According to the CDS of TaGRF2-7A, the primers with SpeI restriction site were designed for cloning the full length cDNA of TaGRF2-7A and construction the expression vector pCAMBIA1304-TaGRF2-7A (Table S1). The pre-miR396b and CDS of TaGRF2-7A sequences were amplified using the genomic DNA and cDNA of Guomai 301 respectively. The pre-miR396b and CDS of TaGRF2-7A were constructed into the expression vector pCAMBIA1304 and formed pCAMBIA1304-35S::pre-miR396b and pCAMBIA1304-35S::TaGRF2-7A. The pCAMBIA1304-35S::pre-miR396b and pCAMBIA1304-35S::TaGRF2-7A were transformed into GV3101 strain (Agrobacterium tumefaciens). 5-week-old tobacco (Nicotiana benthamiana) leaves were infiltrated with pCAMBIA1304-35S::TaGRF2-7A (P1) or pCAMBIA1304 (P3) alone, and co-transformed with pCAMBIA1304-35S::pre-miR396b and pCAMBIA1304-35S::TaGRF2-7A (P2). Tobacco leaves with different treatments at 2, 24, 48, and 72 h post-infiltration were sampled respectively. The dynamic changes of TaGRF2-7A expression were analyzed by qRT-PCR. Actin gene was used as an internal control.

GUS staining

Tobacco was transformed with Agrobacterium carrying plasmid pCAMBIA1304-35S::TaGRF2-7A, and co-transformed with Agrobacterium carrying plasmid pCAMBIA1304-35S::TaGRF2-7A and Agrobacterium carrying plasmid pCAMBIA1304-35S::pre-miR396b separately. The tobacco leaves at 48 h after transformation with the expression vectors were sampled and stained with GUS staining buffer at 37 °C overnight. The stained tobacco leaves were then placed in 70% alcohol in a 60 °C water bath for 10 min, and then the tobacco leaves were placed in anhydrous ethanol in a 60 °C water bath until the tobacco leaves were completely decolorized (Yang, Poretska & Sieberer, 2018).

Results

Spike phenotypes of WT, ptsd1, drs, and ass

The agronomic traits between WT and the mutants ptsd1, drs, and ass were compared. There was no significant difference in plant height between WT and ptsd1 at grain filling stage. However, there were significant differences in spike component traits (Jiao et al., 2019). Compared to WT, the spike length was significantly shorter and spikelets per spike were fewer in ptsd1 (Figs. 1A and 1E; Table 1). The upper spikelets of ptsd1 stopped differentiating, but the basal spikelets could still differentiate at the floret primordium visible stage and the two awns/spikelet reaching apical meristem of the spikelet stage (Figs. 1B and 1C). At the green anther stage, the upper spikelets could not form stamens due to the stopped spikelet differentiation, resulting in a significantly fewer fertile spikelets than that of WT (Fig. 1D). At the grain filling stage, the plant height and spike length were significantly shorter in mutant drs than that in WT, but the spikes of drs were wider (Figs. 1A and 1E). At the floret primordium visible stage, the bottom of the spike was significantly wider than the top of the spike (Fig. 1B). There was already a clear difference in spike length between WT and drs at the two awns/spikelet reaching apical meristem of the spikelet stage (Fig. 1C). At the green anther stage, the spike length was significantly shorter than that of WT, and finally the whole spike was oval (Fig. 1D). At the grain filling stage, the mutant ass was slightly shorter than that of WT in plant height, and the fertile spikelets were significantly fewer than that of WT due to the pistil and stamen degenerations of the apical spikelets (Figs. 1A and 1E; Table 1). At the floret primordium visible stage and the two awns/spikelet reaching apical meristem of the spikelet stage, the apical spikelets of ass developed normally, and there was no significant difference between ass and WT (Figs. 1B and 1C). However, at the green anther stage, the apical spikelets of ass have developed abnormally, and finally the apical spikelets could not developed and set grain normally (Fig. 1D). Wheat spike developmental stages were described as following (Fig. 2; Vahamidis et al., 2014).

Figure 1 Comparison of plant and spike morphology between WT and mutant ptsd1, drs, and ass (left—right).

(A) The plants of WT, ptsd1, drs, and ass at the grain filling stage (left—right). (B) The spikelets of WT, ptsd1, drs, and ass at the floret primordium visible stage. (C) The spikelets of WT, ptsd1, drs, and ass at the two awns/spikelet reaching apical meristem of the spikelet stage. (D) The spikelets of WT, ptsd1, drs, and ass at the green anther stage; (E) The spikes of WT, ptsd1, drs, and ass at grain filling stage. (A) Scale bar = 10 cm; (B) and (C) Scale bar = 1 mm; (D) and (E) Scale bar = 1 cm. The developmental stages were defined referred to that described by Vahamidis et al. (2014).

Table 1 Comparison of agronomic traits between the WT and mutants ptsd1, drs and ass.

Tract	WT	drs	ass	ptsd1	
Plant height/cm	64.45 ± 3.28	38.52 ± 3.15**	53.65 ± 3.34*	50.75 ± 3.09**	
Tiller number	20.73 ± 3.20	23.82 ± 2.20	16.77 ± 2.32	21.73 ± 2.30	
Spike length/cm	11.47 ± 0.53	4.25 ± 0.88**	12.13 ± 0.67	3.86 ± 1.30**	
Spike width/cm	2.11 ± 0.13	2.39 ± 0.22**	1.76 ± 0.21*	1.48 ± 0.10**	
Spikelet number	21.00 ± 2.00	17 ± 2.00*	21.00 ± 2.00	4.50 ± 1.50**	
Fertile spikelet number	20.00 ± 1.00	17 ± 2.00*	12 ± 2.00**	4.50 ± 1.50**	
Notes:

Phenotypic performance of each trait was expressed as means ± standard deviation (n = 10).

** and *, significant difference at P < 0.01 and P < 0.05 by t-test, respectively.

Figure 2 Illustration of the proposed scales of wheat spike differentiation of WT.

(A) Glume primordium visible. (B) Floret primordium visible stage. (C) Terminal spikelet stage. (D) Two awns/spikelet reaching apical meristem of the spikelet stage. (E) Two basal florets fully covered by lemmas stage. (F) Spikelet apical meristem just visible or white anther stage. (G) Green anther stage. (H) Grain filling stage. (A–D) Scale bar = 1 mm. (E–H) Scale bar = 1 cm. The developmental stages were defined referred to that described by Vahamidis et al. (2014).

Tae-miR396b was located on the long arms of chromosomes 6A, 6B and 6D

To explore the evolution of tae-miR396b, the sequence alignment of tae-miR396b with the homologs in other species were carried out. The result indicated that the miR396bs of wheat (T. aestivum L.), Arabidopsis (Arabidopsis thaliana), rice (Oryza sativa), soybean (Glycine max), tobacco (Nicotiana tabacum), Arabidopsis lyrata, and maize (Zea mays) were highly conserved (Fig. 3A). Mature miRNA is incorporated into the miRNA-induced silencing complex (miRISC). As part of miRISC, miRNAs base-pairs target miRNAs and induce their translational repression or deadenylation and degradation. Sequence exploration of miR396b in wheat genome indicated that there were three homeologous genes of tae-miR396b on the long arms of chromosomes 6A, 6B and 6D (Fig. 3B), and they were named as tae-miR396b-6AL, tae-miR396b-6BL, and tae-miR396b-6DL respectively. The sequences of the three homeologous genes were highly conserved except for some SNPs and InDels among them. However, they encoded the same mature tae-miR396b (Fig. 3C).

Figure 3 Characteristic information of tae-miR396b.

(A) Sequence alignment of the mature miR396s in wheat (T. aestivum L.; tae-), Arabidopsis (A. thaliana; ath-), rice (O. sativa; osa-), soybean (G. max; gma-), tobacco (N. tabacum; nta-), Arabidopsis lyrata (A. lyrata; aly-) and maize (Z. mays; zma-). (B) Stem-loop structures of tae-MIR396b-6AL, tae-MIR396b-6BL and tae-MIR396b-6DL. The red part is mature sequence. The yellow part is the stem-loop sequence. The purple part is star sequence. (C) Sequence alignment of the three tae-MIR396b genes. The places marked with light blue are different sequences (SNPs and InDels).

Twenty-three TaGRFs had binding sites of tae-miR396b

To explore how many genes were the targets of tae-miR396b, the binding sites of tae-miR396b were screened in all wheat genes. A total of 203 target genes of tae-miR396b were identified in wheat genome (Table S2). Among the target genes, 23 genes belonged to the GRF gene family. Wheat GRF gene family has 30 genes (Zhang et al., 2021). The sequence alignment of the 30 TaGRFs with tae-miR396b (tae-miR396b: 5′-UUCCACAGCUUUCUUGAACUU-3′) further confirmed that the 23 TaGRFs had the binding sites of tae-miR396b (Fig. 4). This result demonstrated that the most TaGRF genes were the potential targets of tae-miR396b.

Figure 4 Sequence alignment of tae-miR396b with complementary sequences in TaGRF genes.

TaGRFs were highly expressed in spikes

The expressions of the TaGRFs in roots, leaves and spikes of Chinese Spring at GS32 and GS43 stage were analyzed (Fig. 5 and Table S3). The results showed that TaGRF1 (6A, 6B, 6D), TaGRF5 (4A, 7A, 7D) and TaGRF6 (4A, 4B, 4D) were consistently expressed at high levels during root development. TaGRF2 (7A, 7B, 7D), TaGRF3 (2A, 2D), TaGRF4 (2B, 4A, 4B, 4D) and TaGRF10 (6A, 6B, 6D) were lowly expressed or even couldn’t be detected during root development. The expression levels of the 23 TaGRFs were low or even not expressed in leaves. However, during spike development, all the TaGRFs expressed at a high level at GS32, but the expression levels decreased or some TaGRFs were even not expressed at GS43. The above results indicated that TaGRFs were highly expressed in the process of spike development, and the expression levels varied greatly in different developmental stages. Because wheat spikes were at early differentiation stages at GS32, completed differentiation at GS43, TaGRFs played more important roles in spike differentiation.

Figure 5 Expression levels of the TaGRF genes at 2nd node detectable (GS32) and boots just visibly swollen (GS43).

V1, Roots of Chinese Spring at GS32; V2, leaves of Chinese Spring at GS32; V3, spikes of Chinese Spring at GS32; R1, roots of Chinese Spring at GS43; R2, leaves of Chinese Spring at GS43; R3, spikes of Chinese Spring at GS43.

Functional classification of the TaGRFs in GO

Gene Ontology annotation analysis of TaGRFs showed that all TaGRFs played the same functions in molecular functions, biological processes and cellular components (Table 2). They were mainly involved in ATP binding (GO:0005524), regulation of DNA-templated transcription (GO:0006355), and developmental processes (GO:0032502).

Table 2 Functional classification of the TaGRFs in GO.

	Gene ID	+ no	Molecular function	Cellular component	Biological process	
TaGRF1-6A	TraesCS6A01G335900	5	GO:0005524	GO:0005634	GO:0006351	GO:0006355	GO:0032502	
TaGRF1-6B	TraesCS6B01G366700	5	GO:0005524	GO:0005634	GO:0006351	GO:0006355	GO:0032502	
TaGRF1-6D	TraesCS6D01G315700	5	GO:0005524	GO:0005634	GO:0006351	GO:0006355	GO:0032502	
TaGRF2-7A	TraesCS7A01G165600	5	GO:0005524	GO:0005634	GO:0006351	GO:0006355	GO:0032502	
TaGRF2-7B	TraesCS7B01G070200	5	GO:0005524	GO:0005634	GO:0006351	GO:0006355	GO:0032502	
TaGRF2-7D	TraesCS7D01G166400	5	GO:0005524	GO:0005634	GO:0006351	GO:0006355	GO:0032502	
TaGRF3-2A	TraesCS2A01G435100	5	GO:0005524	GO:0005634	GO:0006351	GO:0006355	GO:0032502	
TaGRF3-2D	TraesCS2D01G435200	5	GO:0005524	GO:0005634	GO:0006351	GO:0006355	GO:0032502	
TaGRF4-2B	TraesCS2B01G458400	5	GO:0005524	GO:0005634	GO:0006351	GO:0006355	GO:0032502	
TaGRF4-6A	TraesCS6A01G269600	5	GO:0005524	GO:0005634	GO:0006351	GO:0006355	GO:0032502	
TaGRF4-6B	TraesCS6B01G296900	5	GO:0005524	GO:0005634	GO:0006351	GO:0006355	GO:0032502	
TaGRF4-6D	TraesCS6D01G245300	5	GO:0005524	GO:0005634	GO:0006351	GO:0006355	GO:0032502	
TaGRF5-4A	TraesCS4A01G434900	5	GO:0005524	GO:0005634	GO:0006351	GO:0006355	GO:0032502	
TaGRF5-7A	TraesCS7A01G049100	5	GO:0005524	GO:0005634	GO:0006351	GO:0006355	GO:0032502	
TaGRF5-7D	TraesCS7D01G044200	5	GO:0005524	GO:0005634	GO:0006351	GO:0006355	GO:0032502	
TaGRF6-4A	TraesCS4A01G255000	5	GO:0005524	GO:0005634	GO:0006351	GO:0006355	GO:0032502	
TaGRF6-4B	TraesCS4B01G060000	5	GO:0005524	GO:0005634	GO:0006351	GO:0006355	GO:0032502	
TaGRF6-4D	TraesCS4D01G059600	5	GO:0005524	GO:0005634	GO:0006351	GO:0006355	GO:0032502	
TaGRF9-4A	TraesCS4A01G291500	5	GO:0005524	GO:0005634	GO:0006351	GO:0006355	GO:0032502	
TaGRF9-4D	TraesCS4D01G020300	5	GO:0005524	GO:0005634	GO:0006351	GO:0006355	GO:0032502	
TaGRF10-6A	TraesCS6A01G257600	5	GO:0005524	GO:0005634	GO:0006351	GO:0006355	GO:0032502	
TaGRF10-6B	TraesCS6B01G267500	5	GO:0005524	GO:0005634	GO:0006351	GO:0006355	GO:0032502	
TaGRF10-6D	TraesCS6D01G238900	5	GO:0005524	GO:0005634	GO:0006351	GO:0006355	GO:0032502	
Note:

ATP binding (GO:0005524); nucleus (GO:0005634); DNA-templated transcription (GO:0006351); regulation of DNA-templated transcription (GO:0006355); developmental process (GO:0032502).

Expression profiles of tae- miR396b and TaGRFs in WT, drs, ass, and ptsd1

The qRT-PCR was performed to analyze the expression profiles of tae-miR396b and TaGRFs in spikelets of WT and mutants drs, ass, and ptsd1 at three spikelet developmental stages. Expression profiles of tae-miR396b and TaGRFs in the spikes of WT and mutants were different at the floret primordium visible stage, the two awns/spikelet reaching apical meristem of the spikelet stage, and the green anther stage. The relative expression levels of tae-miR396b and TaGRFs were also changed at different stages. Generally, the expression patterns of tae-miR396b and most TaGRFs were similar, but their expression levels were significantly different (Figs. 6 and 7).

Figure 6 Expression analysis of tae-miR396b and 11 TaGRF genes at different stages by qRT-PCR.

(A–L) The qRT-PCR results of tae-miR396b and 11 TaGRF genes in WT, ptsd1, drs, and ass at three different stages. S1, floret primordium visible stage; S2, two awns/spikelet reaching apical meristem of the spikelet stage; S3, green anther stage. The internal reference of tae-miR396b was U6 gene, and the internal reference of 11 TaGRF genes were actin gene. Vertical bars indicated standard deviation. Bars (means) with different letters are significantly different (P < 0.01).

Figure 7 Expression analysis of 12 TaGRF genes at different stages by qRT-PCR.

(A–L) The qRT-PCR results of 12 TaGRFs genes in WT, ptsd1, drs, and ass at three different stages. S1, floret primordium visible stage; S2, two awns/spikelet reaching apical meristem of the spikelet stage; S3, green anther stage. Data were normalized to actin gene, and vertical bars indicated standard deviation. Bars (means) with different letters are significantly different (P < 0.01).

The real-time quantitative analysis revealed that the expression of tae-miR396b initially increased and then decreased in WT and ptsd1, while little change in expression in drs and ass. Notably, the expression level of tae-miR396b was significantly higher in WT compared to mutants drs and ass. In WT, the expression levels of tae-miR396b were significantly different at the three stages. Most TaGRFs were highly expressed at S1, decreased at S2 and then increased again at S3, especially TaGRF1 (6A, 6B, 6D), TaGRF2 (7A, 7B, 7D), TaGRF3 (2A, 2D), TaGRF4-2B and TaGRF6-4A. The expression levels of TaGRF1 (6A, 6B, 6D), TaGRF2 (7A, 7D), TaGRF3-2D, TaGRF4-6D, TaGRF6 (4A, 4B, 4D) and TaGRF9 (4A, 4D) were all significantly increased at S3, implying they played roles in regulating spike development. The expression levels of TaGRF1 (6A, 6D), TaGRF2 (7A, 7B, 7D), TaGRF4 (6A, 6B), TaGRF5 (4A, 7A, 7D), and TaGRF10-6 (A. B, D) between WT and the mutants were significantly different at S1, S2 and S3, implying they might be the causes of abnormal spike differentiation in the mutants. In summary, the expression levels of most TaGRFs were opposite to that of tae-miR396b, which demonstrated that the expressions of most TaGRFs were negatively regulated by tae-miR396b.

Tae-miR396b had different regulatory effects on TaGRFs in WT and mutants drs, ass and ptsd1

In WT, the expression of tae-miR396b was negatively correlated with the expressions of TaGRF1 (6A, 6B, 6D), TaGRF2 (7A, 7B, 7D) and TaGRF3 (2A, 2D) (Fig. 8). In drs, the expression of tae-miR396b was significantly negatively correlated with the expressions of 7 TaGRFs including TaGRF3-4D, TaGRF4-6D, TaGRF5-4A, TaGRF4 (6B, 6D), TaGRF9-4A and TaGRF10-6A (Fig. 9). In ass, the expression of tae-miR396b was negatively correlated with the expressions of most TaGRFs including TaGRF1 (6A, 6B, 6D), TaGRF2 (7A, 7B), TaGRF3-2D, TaGRF4-6D, TaGRF6 (4B, 4D) and TaGRF9 (4A, 4D) (Fig. 10). In ptsd1, the expression of tae-miR396b was negatively correlated with the expressions of TaGRF1 (6A, 6B, 6D), TaGRF2-7D, TaGRF4 (2B, 6D), TaGRF6 (4D, 4D), TaGRF9 (4A, 4D) and TaGRF10-6D (Fig. 11). These results demonstrated that tae-miR396b had different regulatory effects on TaGRFs in WT and the mutants. The expression patterns of TaGRF genes were complex. Both tae-miR396b and TaGRF genes impact mutant spike development to some extent.

Figure 8 Heat map of the correlations between tae-miR396b and TaGRFs expressions in WT.

The color scale indicates the values of Pearson’s rank correlation coefficient. Orange indicates a negative correlation; green indicates a positive correlation. The numbers in the figure indicate the correlation coefficients.

Figure 9 Heat map of the correlations between tae-miR396b and TaGRFs expressions in drs.

The color scale indicates the values of Pearson’s rank correlation coefficient. Orange indicates a negative correlation; green indicates a positive correlation. The numbers in the figure indicate the correlation coefficients.

Figure 10 Heat map of the correlations between tae-miR396b and TaGRFs expressions in ass.

The color scale indicates the values of Pearson’s rank correlation coefficient. Orange indicates a negative correlation; green indicates a positive correlation. The numbers in the figure indicate the correlation coefficients.

Figure 11 Heat map of the correlations between tae-miR396b and TaGRFs expressions in ptsd1.

The color scale indicates the values of Pearson’s rank correlation coefficient. Orange indicates a negative correlation; green indicates a positive correlation. The numbers in the figure indicate the correlation coefficients.

Correlation analysis showed that tae-miR396b was negatively correlated with most TaGRFs, however, it was positively correlated with some TaGRFs, especially in mutant drs. The reason might be that the expressions of TaGRFs were not only regulated by tae-miR396b, but also regulated by other factors, because there were many cis-elements in the promoters of TaGRFs. Obviously, the expression regulations of TaGRF genes were more complex.

TaGRF2-7A was negatively regulated by tae-miR396b

The results of molecular reaction assay in tobacco showed that the expression level of TaGRF2-7A was continuously increased after transformation only with pCAMBIA1304-35S::TaGRF2-7A. When transformed with vector pCAMBIA1304 alone (P3), the transcripts of TaGRF2-7A was not detected. However, the expression level of TaGRF2-7A was very low when co-transformed with pCAMBIA1304-35S::pre-miR396b and pCAMBIA1304-35S::TaGRF2-7A (Fig. 12). This result demonstrated that tae-miR396b could interact with the transcripts of TaGRF2-7A, and negatively regulated the expression of TaGRF2-7A.

Figure 12 Expression levels of TaGRF2-7A at different time points after transformation.

(A) Histogram of expression levels of TaGRF2-7A at different time points after transformation. (B) Line graph of expression levels of TaGRF2-7A at different time points after transformation. (P1, Tobacco transformed with pCAMBIA1304 -35S::TaGRF2-7A alone, P2, co-transformed with pCAMBIA1304-35S::pre-miR396b and pCAMBIA1304-35S::TaGRF2-7A). Vertical bars represent standard deviation. Asterisks indicate significant difference or highly significant difference between P1 and P2. An asterisk (*) and two asterisks (**) indicate significant difference (P < 0.05) and highly significant difference (P < 0.01) using Student’s t-test, respectively.

The regulatory effect of tae-miR396b on TaGRF2-7A analyzed by GUS staining

GUS staining result indicated that the tobacco leaf transformed with only pCAMBIA1304-35S::TaGRF2-7A was stained deeply, indicating accumulation of the transcripts of TaGRF2-7A. Oppositely, the tobacco leaf transformed with both pCAMBIA1304-35S::TaGRF2-7A and pCAMBIA1304-35S::pre-miR396b could not be stained, indicating the transcripts of TaGRF2-7A were digested (Fig. 13). This result demonstrated that TaGRF2-7A was a target of tae-miR396b. The expression of TaGRF2-7A was negatively regulated by tae-miR396b.

Figure 13 TaGRF2-7A regulated by tae-miR396b was verified by GUS chemical staining in tobacco leaf.

Left side of tobacco leaf: transformed with pCAMBIA1304-35S::TaGRF2-7A alone; Right side of tobacco leaf: co-transformed with pCAMBIA1304-35S::pre-miR396b and pCAMBIA1304-35S::TaGRF2-7A.

Discussion

Characteristics and functions of tae-miR396b and TaGRFs

Tae-miR396b has three homeologous genes located on the long arms of chromosomes 6A, 6B, and 6D, with some SNPs and Indels among them (Fig. 3B). SNP markers obtained from molecular genetics and genomics provide broad prospects for wheat breeding (Collard & Mackill, 2007). The SNPs and Indels can be used to distinguish the three homeologous genes of tae-miR396b, and used in wheat molecular design breeding.

Plant GRF proteins typically have two conserved domains at the N-terminus: QLQ and WRC. QLQ domains interact with GRF interactors (GIFs) to form transcriptional activators involved in biological processes of plant growth and development (Kim & Kende, 2004). The expression levels of GRFs are usually higher in actively growing tissues than in mature tissues (Horiguchi, Kim & Tsukaya, 2005; Kim, Choi & Kende, 2003; Rodriguez et al., 2010). Wheat GRF gene family have 30 members. TaGRFs have a large number of cis-regulatory elements related to growth and development, hormones and stress (Zhang et al., 2021). Gene ontology annotation showed that TaGRFs were mainly involved in ATP binding (GO:0005524), regulation of DNA templated transcription (GO: 0006355) and developmental processes (GO:0032502) (Table 2). These indicated that TaGRFs also played important roles in wheat development. A total of 23 TaGRFs had the target sites of tae-miR396b (Fig. 4), indicating their expressions were regulated by tae-miR396b. These suggested that tae-miR396b and TaGRFs played important roles in wheat growth and development.

Tae-miR396b and TaGRFs were involved in the regulation of wheat spike development

More and more studies have shown that miR396 is involved in developments of plant leaves, roots, florets and grains by regulating GRFs (Bazin et al., 2013; Liu et al., 2009; Liu et al., 2014; Yu et al., 2020). The regulation mechanism of miR396 on crop growth is complex, and it can offset the interference caused by external changes through autonomous regulation. The miR396-mediated GRF regulation (miR396-GRF) module has been demonstrated as a potential application in improving plant biomass, crop yield, stress tolerance, and increasing the efficiency of genetic transformation in plants (Cao et al., 2007; Chen et al., 2019; Kong et al., 2020). Overexpression of miR396-resistant GRF1 do not increase leaf size, because miR396a can control maize leaf growth by finely controlling transcript levels through a series of complex processes to counteract the effects of overexpression (Nelissen et al., 2015). Tissue specific expression analysis of TaGRFs indicated that the most TaGRFs were expressed in both roots and leaves, but all TaGRFs expressed at high levels in young spikes (Fig. 5). The results of qRT-PCR showed that the expression levels of TaGRFs were varied at different spike developmental stages (Figs. 6 and 7). These suggested that TaGRFs were involved in the regulation of wheat spike development.

Tae-miR396b and TaGRFs were involved in the abnormal spike differentiation of mutants drs, ass, and ptsd1

GRFs are involved in the formation of the reproductive organs of A. thaliana, affecting the differentiation of female and stamen, and the formation of embryo sac. It is of great significance for the reproductive capacity and generational continuity of A. thaliana (Lee et al., 2014). In maize, 14 ZmGRF genes have been identified, ZmGRF2 and ZmGRF11 may be involved in the growth and development of spike formation and contribute to the improvement of plant yield (Zhang et al., 2008). Rice GRFs are involved in floral organ development (Li et al., 2010; Luo et al., 2005); overexpression of OsmiR396d targeted inhibition of OsGRF6 leads to dysplasia of flower organs (Liu et al., 2014); both OsGRF6 and OsGRF10 can bind the promoters of OsCR4 and OsJMJ706, which influence rice fertility and floral organ development (Pu et al., 2012; Sun & Zhou, 2008). The mutants had the same genetic background with WT. Genetic study demonstrated that a single or two Mendelian genes controlled the spike phenotypes of the mutants. The expression levels of tae-miR396b and TaGRFs in spikes of WT and mutants drs, ass and ptsd1 at different stages were significantly different (Figs. 6 and 7). The most significantly differentially expressed genes were TaGRF1-6A, TaGRF2-7A, TaGRF4 (6A, 6B, 6D), TaGRF5-7A, TaGRF6 (4A, 4D) and TaGRF9 (4A, 4D). In Arabidopsis, lines with GRF3 mutations in the miR396 binding site and lines overexpressing AtGRF5 can effectively delay leaf senescence (Debernardi et al., 2014). GRFs are also involved in the growth and development of plant roots, stems (Kim, Choi & Kende, 2003), and flowers (Liang et al., 2013). We speculated that these TaGRFs with significantly different expression levels played important roles in spike development. The result suggested that the tae-miR396b-TaGRFs module was involved in the abnormal spike development of the mutants. Correlation analysis showed that tae-miR396b was positively or negatively correlated with TaGRFs (Figs. 8–11). The positive correlations were contradictive with the regulatory relationships predicted between Tae-miR396b and TaGRFs. This result suggested that the expressions of TaGRFs were regulated by a complex system, not only by tae-miR396b. The expression of TaGRF2-7A could be inhibited by tae-miR396b directly (Figs. 12 and 13), but whether all the other TaGRFs were regulated directly by tae-miR396b still needed to be verified.

Conclusions

Three homeologous genes of tae-miR396b are on the long arms of chromosome 6A, 6B and 6D with some SNPs and Indels. A total of 23 TaGRFs have the target sites of tae-miR396b, and they express at the highest levels in wheat spikes. Tae-miR396b and TaGRFs express at significantly different levels in spikes of Guomai 301 and mutants drs, ass and ptsd1 at the floret primordium visible stage, the female and male primordium differentiation stage, and the green anther stage. The most TaGRFs are negatively regulated by tae-miR396b, which have different regulatory effects on TaGRFs in WT and the three spike mutants. The expression of TaGRF2-7A is directly negatively regulated by tae-miR396b. The expressions of TaGRFs are regulated by a complex system, not only by tae-miR396b. Tae-miR396b and TaGRFs are involved in the regulation of wheat spike development, and play important roles in the abnormal spike development of the mutants. This study has identified tae-miR396 and its important potential target GRF family gene members, revealed the roles of tae-miR396b-TaGRFs module in spike differentiation of wheat, and laid a foundation for further research on the roles of tae-miR396b and TaGRFs in spike development of wheat. It has provided candidate functional genes for wheat breeding, especially for spike improvement.

Supplemental Information

Supplemental Information 1 The DNA sequences of the primers used for genes expression analysis.

Supplemental Information 2 Target gene prediction results of tae-miR396b.

Supplemental Information 3 The transcripts per million (TPM) values of the TaGRFs in roots, leaves and spikes of Chinese Spring at vegetative and reproductive stages.

Supplemental Information 4 The raw data of qRT-PCR.

We are grateful for the assistance by Shangqiu Academy of Agricultural and Forestry Sciences. We thank National Centre of Engineering and Technological Research of Wheat for the technical support for the cultivations. We thank all the members for their work on this study.

Additional Information and Declarations

Competing Interests

Author Contributions

Data Availability

The authors declare that they have no competing interests.

Ziping Yao conceived and designed the experiments, performed the experiments, prepared figures and/or tables, and approved the final draft.

Qi Wang conceived and designed the experiments, performed the experiments, prepared figures and/or tables, and approved the final draft.

Ying Xue conceived and designed the experiments, performed the experiments, prepared figures and/or tables, and approved the final draft.

Zhiheng Liang conceived and designed the experiments, prepared figures and/or tables, and approved the final draft.

Yongjing Ni analyzed the data, prepared figures and/or tables, authored or reviewed drafts of the article, and approved the final draft.

Yumei Jiang analyzed the data, prepared figures and/or tables, authored or reviewed drafts of the article, and approved the final draft.

Peipei Zhang performed the experiments, analyzed the data, prepared figures and/or tables, and approved the final draft.

Ting Wang performed the experiments, analyzed the data, prepared figures and/or tables, and approved the final draft.

Qiaoyun Li analyzed the data, authored or reviewed drafts of the article, and approved the final draft.

Lei Li analyzed the data, authored or reviewed drafts of the article, and approved the final draft.

Jishan Niu analyzed the data, authored or reviewed drafts of the article, and approved the final draft.

The following information was supplied regarding data availability:

The raw data of qRT-PCR are available in the Supplemental Files.

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
