# Peer review of "Tae-miR396b regulates TaGRFs in spikes of three wheat spike mutants"

_PeerJ, doi:10.7717/peerj.18550_

## Round 0.1 · original submission · Major Revisions

It is my opinion as the Academic Editor for your article - Tae-miR396b regulates TaGRFs in spikes of three wheat spike mutants - that it requires a number of Major Revisions, but we will be willing to receive a revised version of the manuscript. The reviewer comments are shown below. Please carefully address all the issues and resubmit.

Fabio Nogueira
Academic Editor

**Language Note:** PeerJ staff have identified that the English language needs to be improved. When you prepare your next revision, please either (i) have a colleague who is proficient in English and familiar with the subject matter review your manuscript, or (ii) contact a professional editing service to review your manuscript. PeerJ can provide language editing services - you can contact us at [email protected] for pricing (be sure to provide your manuscript number and title). – PeerJ Staff

·

Basic reporting

The english is clear and well structured. The cited literature offers the information that the document needs, however, it deserves more literature to complement the information especially in the introduction and discussion. The document lacks a defined hypothesis. Methods were described with sufficient detail and information. Results are well structured but lacks a lineal connection with the whole document.

Experimental design

Regarding the results,

From lines 125 to 142, the description about the spike phenotypes in WT and mutants was clear and Figure 1 clearly illustrated the information given into the results; however, I recommend to put the names of the mutants on the images in every picture of the figure to better connect the information. Also, I strictly recommend to include a supplementary figure containing an illustrating information about the stages of spike/spikelet development in WT to have a better understanding of the results presented.

In line 144 the authors use past tense referring to miR396b conservation (MiR396b was highly conserved among wheat). They must use present tense. Also, I would like to know what is the connection between the position into the chromosomes and the function.

Although the authors mentioned the predicted number of tae-miR396b target genes and the GRF genes containing binding sites of tae-miR396b, in lines 152 to 155 , they did not connect these results with the main part of the research. It is not only necessary to mention results but also to connect them with the whole document.

The authors have shown the expressions of the TaGRFs in roots, leaves and spikes (Linnes 157 to 166). I did not understand why they showed the GRF expression in roots and leaves; it does not seem to have a relationship with the main goal of the work. In addition, the authors did not show the miR396b expression to complement the gene expression profile of the tae-miR396b/GRF pathway. It is necessary to connect each other in order to show that miR396b probably regulates the GRF expression in spikes. It was not mentioned how the gene expression of the GRF genes was evaluated neither in the text nor in the figure.

There is a huge amount of information from the real-time quantitative expression analysis; however, the information provided by the authors in lines 173 to 177 is very slight. In addition, between lines 178 to 188 the authors mention the results as observed in Figure 5 and 6 but they neither connect to the main part of the research nor generate possible hypotheses derived from these analyzes.

In lines 190 to 198, the authors again mention the results without offering a connection with the aim of the work.

In lines 204 to 209 the authors conclude that TaGRF2-7A is negatively regulated by tae-miR396b. I am afraid that what they did, as shown, is not enough to conclude this; thus, additional experiments could help to answer this question.

In Figure 11, the authors show the relative expression of the aGRF2-7A from tobacco leaves infiltrated with 35S::TaGRF2-7A and 35S::pre-miR396b. It would be interesting if they use these vectors to transform wheat in order to see the phenotypic effects on spikes.

Validity of the findings

Complementing the information available in literature, generating new hypothesis with the results, and doing additional eperiments to validate the main conclusion can help to complement the research.

Additional comments

Although there is no novelty regarding the effect of the miR396 in spikes, the interaction between this microRNA and the mutants drs, ass and ptsd1 constitutes the central point of the work that needs to be clarified with more details.

Reviewer 2 ·

Basic reporting

This manuscript describes the expression pattern of tae-miR396b and TaGRFs in wheat. The authors previously identified three spike-shaped mutants (drs, ass, ptsd1) from the EMS-induced population. In this study, they compared expression levels of the genes using those mutants.

My major concern is that the data do not support the conclusion of the results. For example, in the abstract, the authors mention "The expression level of tae-miR396b in WT was significantly higher than that in mutants drs, ass and ptsd1 at S1 and S2."; however, I don't see significant differences in ptsd1 mutant in Figure 5A. There is no description of the statistical test. "The most TaGRFs were negatively regulated by tae-miR396b." Which data supports this statement?

In the results "Spike phenotypes of WT, ptsd1, drs, and ass". Is this result novel or already published elsewhere? If it is not new, please delete this section.

Experimental design

Figure 4: What does it mean "V3: spikes of Chinese Spring at vegetative stage"? If RNA was extracted from spikes, it indicates the stage is reproductive.

Figures 5 and 5 look highly related, why separated?

Validity of the findings

As I mentioned above, the conclusions are not clear. What was the key finding from the expression analysis?

Additional comments

The terminology of the spike developing stage should be corrected.
floret primordium formation stage > floret primordium stage
female and male primordium differentiation stage > female and male primordium stage

Reviewer 3 ·

Basic reporting

Please see the PDF for details briefly below:
Figures and Figure Legends: Ensure high-resolution images and clear legends that succinctly explain what the figure shows, including specifying what readers should notice in each panel. Consistency in font size, style, and color coding across all figures will improve the professional appearance of the manuscript.
Clarity and Language: Some sentences are quite long and contain multiple pieces of information, suggesting a need for breaking these into shorter sentences to improve readability. Ensure that all acronyms are explained when first introduced.
References: There's a need to ensure that all references are correctly backing up the sentences where they have been mentioned, pointing out any discrepancies or misattributions.

Experimental design

Please see the PDF for details briefly below:
Methodological Clarity: The Methods section requires precision in describing methodologies for better clarity and reproducibility. This includes specifying the EMS concentration used for seed treatment, stating the version of the databases used, and defining any thresholds or parameters set within analyses.
Experimental Validation: Additional experiments such as functional validation of TaGRFs using CRISPR/Cas9 or RNAi technology, and comprehensive transcriptomic analysis could provide more depth to the study.

Validity of the findings

Please see the PDF for details briefly below:
Statistical Analysis: The Results section could be improved by including statistical analysis (e.g., ANOVA) to compare the expression levels across all stages and conditions, and presenting individual data on the graphs as dots to solidify observations.
Data Interpretation: Ensure that findings are discussed in light of current literature, highlighting the novel aspect of the study compared to existing work. Exploring how the results could influence future breeding strategies for yield improvement would provide practical implications of the research.

Additional comments

Please see the PDF for details briefly below:
Discussion of Unaddressed Topics: The Discussion should explore the potential evolutionary advantages or functional diversification conferred by SNPs and Indels in tae-miR396b, as well as delve deeper into the potential molecular mechanisms by which tae-miR396b regulates TaGRFs. It should also compare the miR396-GRF pathway's role in spike development between wheat and other cereal crops, providing a comparative view that could highlight unique targets for crop improvement.
Agronomic Implications and Future Research Directions: Expanding on how manipulating the tae-miR396b-TaGRF pathway could be harnessed to improve wheat yield, resilience, or quality, and suggesting specific future studies or methodologies that could further elucidate the functions of tae-miR396b and TaGRFs in wheat.

Annotated reviews are not available for download in order to protect the identity of reviewers who chose to remain anonymous.

---

## Round 0.2 · Minor Revisions

Dear Dr. Niu,

We now have received the reviews of three experts on the field. Although the manuscript has improved, it still needs modifications to be considered for publication in PeerJ life & Environment. Particularly, please answer the issues still raised by reviewers#2 and #3.

Looking foward to hearing from you.

Warm regards,
Fabio Nogueira
Academic Editor
PeerJ Life & Environment

·

Basic reporting

The authors have fulfilled most of the requests I have made.

Experimental design

In the box "Validity of the findings", I put all the information

Validity of the findings

Authors include the information into the Figure 1 and, also, include a new figure (Figure 2) that ilustrates the proposed scales of wheat spike diferentiation of WT. These information clearly help to a better compreension during reading.

In line 144 the authors have corrected the grammar tense referring to miR396b conservation. Although they explained that there is no connection between the miR396b position into the chromosomes and the function, they have justified by being and important information for wheat cytogenetics.

The authors have not only complemented the information about the tae-miR396b target genes but also have included additional results that suggest that TaGRF2-7A is regulated by tae-miR396b. Altogether, these information helps to support the main goal of the research.

Initially, the information provided in lines 173 to 188 was presented as a amount of unconected data. Now, the authors have made some modifications, they tried to conect it with the document and, also, sugest possible TaGRFs functions and tae-miR396b targets.

Previously, I had sugested additional experiments to support the conclution that TaGRF2-7A is negatively regulated by tae-miR396b. The authors have done an additional experiment. They have used the GUS reporter to support the data.

Additional comments

The authors did not attend my suggestion about the expressions of the TaGRFs. I do not see, still, the relationship between GRF expression in roots and leaves with the main goal of the work. And the response in the letter does not support the information.

Reviewer 2 ·

Basic reporting

As I previously mentioned, the terminology of the developing spike stage was wrong. The authors need to follow the literature references such as Kirby EJM, Appleyard M (1981) Cereal development guide, 1st edn. Cereal Unit, Kenilworth, Bonnett OT (1966) Inflorescences of maize, wheat, rye, barley and oats: their initiation and development. University of Illinois College of Agriculture, Agricultural Experimental Station, and Zadoks growth scale. Figure 2 must be modified and moved to a supplementary figure as reviewer 1 recommended.
In the Figure 5, V3: spikes of Chinese Spring at vegetative stage was not corrected. "spike" might be stem or shoot or something, otherwise it is not vegetative stage.

Experimental design

I don't understand why the three independent mutants (totally different morphology) were compared. Is there any evidence that these three mutants are sharing genetic mechanisms?

Validity of the findings

Statistical tests are necessary to support their hypothesis (Figure 6 to 12).

Reviewer 3 ·

Basic reporting

The authors have addressed the reviewers' comments well. The language has improved throughout the manuscript, making it clearer. The figures have also improved significantly, and the new versions better support the conclusions of the manuscript. However, a few changes are still necessary before the manuscript can be accepted for publication.

The results section still lacks a brief description and reasoning before diving into the results. At least one or two introductory sentences would be helpful.

In the section titled “Spike phenotypes of WT, ptsd1, drs, and ass,” the authors mention significant changes, but how many replicates were conducted? How do the authors confirm that these changes are significant? A single image is unlikely to be sufficient. Please provide measurements and data analysis to support the claims. Additionally, there is a reliance on descriptive language rather than scientific terminology. Incorporating measurements and statistics would strengthen the data presentation.

Line 168: The statement "This result demonstrated that most TaGRF genes were the targets of tae-miR396b" should be revised to indicate potential targets unless the authors have experimentally validated this claim.

In the results section titled “Expression profiles of tae-miR396b and TaGRFs in WT, drs, ass, and ptsd1,” on Figure 6, please clarify how the expression levels are indicated. Is it log2 fold change, or something else?

The discussion section has improved significantly, and I have no further comments.

Experimental design

The GUS staining procedure requires additional details. Please specify whether the full length of the gene or just the coding sequence was used for the constructs. Additionally, provide details on the optical density (OD) measurements.

For real-time qRT-PCR, how many replicates were performed? Please indicate significant differences in Figure 6, and use appropriate statistical methods to calculate significance. Without statistical analysis, the results remain inconclusive and overly descriptive.

Validity of the findings

In the conclusion section, please revise the statement to indicate that the TaGRFs are “potential” target sites of tae-miR396b, as the current evidence is not strong enough to confirm this definitively.

Additional comments

Throughout the manuscript, ensure that any claims or conclusions drawn from experimental data are adequately supported by statistical analysis and appropriate replicates. Descriptive results without statistical validation may weaken the overall scientific rigor of the study.

Consider revising the introduction to provide a clearer connection between the study's objectives and the broader implications of the findings. This will help contextualize the work and highlight its relevance.

Ensure all figure legends are comprehensive and include enough detail to allow the reader to fully understand the data presented without referring back to the main text. For example, in Figure 6, clarify the method used to represent expression levels (e.g., log2 fold change) and specify whether error bars represent standard deviation or standard error of the mean.

It might be helpful to include a brief methods validation section, especially for qRT-PCR and GUS staining, to reassure readers that the techniques used are accurate and appropriate for the study.

Review the manuscript for consistency in terminology and abbreviations, ensuring that all terms are defined when first used and used consistently throughout.

---

## Round 0.3 · accepted · Accept

Dear Dr. Niu,

Thank you for your submission to PeerJ.

Your manuscript - Tae-miR396b regulates TaGRFs in spikes of three wheat spike mutants - has been Accepted for publication.

Best wishes

Fabio Nogueira